# Black-Hole Models in Loop Quantum Gravity

**Martin Bojowald**

Institute for Gravitation and the Cosmos, The Pennsylvania State University, 104 Davey Lab,
University Park, PA 16802, USA; bojowald@gravity.psu.edu

**Abstract:** Dynamical black-hole scenarios have been developed in loop quantum gravity in various ways, combining results from mini and midisuperspace models. In the past, the underlying geometry of space-time has often been expressed in terms of line elements with metric components that differ from the classical solutions of general relativity, motivated by modified equations of motion and constraints. However, recent results have shown by explicit calculations that most of these constructions violate general covariance and slicing independence. The proposed line elements and black-hole models are therefore ruled out. The only known possibility to escape this sentence is to derive not only modified metric components but also a new space-time structure which is covariant in a generalized sense. Formally, such a derivation is made available by an analysis of the constraints of canonical gravity, which generate deformations of hypersurfaces in space-time, or generalized versions if the constraints are consistently modified. A generic consequence of consistent modifications in effective theories suggested by loop quantum gravity is signature change at high density. Signature change is an important ingredient in long-term models of black holes that aim to determine what might happen after a black hole has evaporated. Because this effect changes the causal structure of space-time, it has crucial implications for black-hole models that have been missed in several older constructions, for instance in models based on bouncing black-hole interiors. Such models are ruled out by signature change even if their underlying space-times are made consistent using generalized covariance. The causal nature of signature change brings in a new internal consistency condition, given by the requirement of deterministic behavior at low curvature. Even a causally disconnected interior transition, opening back up into the former exterior as some kind of astrophysical white hole, is then ruled out. New versions consistent with both generalized covariance and low-curvature determinism are introduced here, showing a remarkable similarity with models developed in other approaches, such as the final-state proposal or the no-transition principle obtained from the gauge-gravity correspondence.

**Keywords:** loop quantum gravity; black holes; quantum corrections; covariance

## 1. Introduction

Black-hole models have recently regained considerable attention in loop quantum gravity. A certain consensus seems to have formed according to which the singularity in a classical black hole is replaced by a non-singular phase in which the density and curvature are not infinite but large, such that infalling matter might bounce back and re-emerge after the initial horizon has evaporated. Such models are not only used in conceptual discussions about a possible non-singular fate of black holes, but even in phenomenological descriptions that aim to derive potentially observable effects from the re-emergence of matter.

It is important to note that none of these models are based on consistent embeddings of possible effects from loop quantum gravity in a covariant space-time theory. Rather, these models assume that effects suggested by some equations of loop quantum cosmology, such as bounded density or

curvature, can be modeled reliably by modified line elements that amend the singular solutions of general relativity in various forms. Therefore, these models implicitly assume, but do not show, that loop quantum gravity has a well-defined semiclassical description in which its dynamics can be described by space-time equipped with Riemannian geometry. Since loop quantum gravity aspires to be a background-independent approach to quantum gravity, however, the structure of space-time cannot be presupposed but should rather be derived from the theory. Current models of black holes therefore have important conceptual lacunae.

Provided models are sufficiently controlled for a space-time analysis to be possible, which in practice means that they have a sufficiently developed canonical structure, general covariance can be tested by various means. Recently, it has been shown by direct calculations that all current loop-inspired black-hole models of bounce form violate general covariance and therefore fail to describe space-time effects in any meaningful way [1–4]. It is important to note that these results were obtained directly within the proposed models. They are therefore independent of any difference between approaches that have been put forward in order to formulate inhomogeneous models of loop quantum gravity, such as hybrid models [5], the dressed-metric approach [6], partial Abelianization in spherically symmetric models [7], or using timelike homogeneous slices in static spherically symmetric space-times [8], just to name those to which the recent no-go results about covariance directly apply.

Fortunately, even before these results became available, it had been found that covariance can be preserved by some loop effects at least in a deformed way, which respects the number of classical symmetries underlying general covariance but may change their algebraic relationships [9–17]. Deformation of the algebra, as opposed to violation of some gauge transformations as found in most of the approaches mentioned in the preceding paragraph, makes sure that the theory remains background independent in the sense that transformations remove the same number of gauge degrees of freedom as in the classical theory. However, as a consequence, space-time is rendered non-Riemannian—unless field redefinitions are applied in certain cases—and at present no universal non-Riemannian space-time geometry has been identified that could describe all the deformation effects.

Such a deformed theory is still predictive in principle because it has a consistent canonical formulation that defines unambiguous observables. However, the relationship between gauge invariance and slicing independence (according to a suitable space-time structure) is less obvious than in the classical limit, complicating interpretations of space-time effects such as black holes. The resulting non-Riemannian structure would have to deviate from standard constructions in very basic ways because it is modified even in properties of generic tensorial objects encoded by the tensor-transformation law (which is replaced by gauge transformations in a canonical formulation). Therefore, deformed geometrical properties cannot be captured completely by well-known classical ingredients such as torsion or non-metricity.

Possible space-time structures consistent with deformed symmetries are still being explored. For instance, it is known that deformations of the form implied by effects from loop quantum gravity are different from modified coordinate transformations found in non-commutative [18] or multifractional geometry [19]. There are some relationships with deformed Poincaré symmetries in suitable limits in which a Minkowski background is obtained [20,21], which confirms that Lorentz transformations are not necessarily violated, but the Minkowski reduction of general space-time transformations is rather strong and cannot be sufficient for a complete understanding of deformed space-time structures. As we will show here, what is known about the resulting consistent space-time structures suggests a markedly different scenario of non-singular black holes, compared with bounce-based black holes.

We will first review current proposals in Section 2 and point out their hidden assumptions and other weaknesses. Since our focus will be on dynamical aspects and space-time structure, we will not discuss results about black-hole entropy in loop quantum gravity. See [22] for a recent review of this topic. Section 3 will then introduce basic aspects of modified space-time structures in spherically symmetric models. General covariance is analyzed in detail in the two main canonical loop-modified approaches [7,8], in which modified line elements have been proposed, and found to be lacking.

However, the former proposal unwittingly hints at an important role played by deformed space-time structures and signature change [1], found independently (and earlier) by direct studies of consistent constraints in spherically symmetric models [10–13].

Signature change is a generic implication of modified space-time structures in models of loop quantum gravity, which can replace classical singularities but may lead to other unwanted implications such as indeterministic behavior even at low curvature. The role of signature change and possible evasions will also be discussed. The condition that physics at low curvature be deterministic then rules out certain black-hole models, including bouncing ones, and suggests new ones that are compatible with determinism as well as generalized covariance. These scenarios show interesting relationships with other proposals unrelated to loop quantum gravity. There is therefore a refreshing contrast with bounce-based black holes in loop quantum gravity, which are often put in opposition to other approaches.

## 2. Proposals

Recent interest in a certain type of black-hole models in loop quantum gravity was rekindled by the discussion in [23] which suggested that "a strong short-scale repulsive force due to quantum effects", motivated by certain claims in loop quantum cosmology, might cause collapsing matter in a black hole to bounce back well before evaporation could have reduced the black hole mass to a tiny Planckian value. Here, we will not be concerned with the question of whether related phenomenological studies are meaningful. We will rather address the more basic conceptual question of whether the proposal, and in particular its claimed relationship with loop quantum cosmology, can be realized in a consistent space-time framework that respects general covariance. Our discussion will also be independent of the dubious claim that loop quantum effects imply a "strong short-scale repulsive force;" see [24].

### 2.1. Basic Premise of Bounce-Based Black Holes

Much of the analysis in [23] is based on a postulated line element

$$ds^2 = -F(r)du^2 + 2dvdr + r^2(d\vartheta^2 + \sin^2\vartheta d\varphi^2) \tag{1}$$

that modifies the classical Schwarzschild solution by the inclusion of an additional term in the function

$$F(r) = 1 - \frac{2mr^2}{r^3 + 2\alpha^2 m} \tag{2}$$

if $\alpha \neq 0$. Furthermore, here, we will not be interested in the question of whether this functional form is justified based on models of loop quantum gravity. A more basic question is whether any modification suggested by models of loop quantum gravity can be compatible with general covariance, such that its dynamical solutions are consistent and can be expressed in terms of a well-defined space-time line element. The specific function (2) was first proposed in [25], where it was also shown that it may be obtained as a solution of a covariant theory, such as general relativity with a suitable stress-energy tensor that falls off quickly with increasing $r$. The question we pose here is more fundamental and works in the opposite direction: starting with a specific modified theory or at least a set of model equations that are not gauge-fixed so as to keep space-time properties accessible, is it possible to express its solutions in the form of Riemannian line elements consistent with gauge transformations.

This question is highly non-trivial because modifications in models of loop quantum cosmology are first implemented in the Hamiltonian constraint of the theory. This constraint generates gauge transformations which, together with the transformations generated by the diffeomorphism constraint, are equivalent to space-time coordinate changes in the classical theory. If the Hamiltonian constraint is modified, gauge transformations may be non-classical, and it is no longer guaranteed that, when applied to metric components, they remain dual to coordinate changes of coordinate differentials

$\mathrm{d}x^a$. If this is not the case, the expression $\mathrm{d}s^2 = g_{ab}\mathrm{d}x^a\mathrm{d}x^b$ is not invariant, and fails to define a meaningful line element on which geometry could be based. We conclude that two non-trivial properties must be shown for any desired space-time effect, such as a bouncing black-hole interior, to be meaningful:

(i)  It must be possible to obtain the effect as a specific solution $g_{ab}$ of a consistent set of field equations.
(ii)  Together with the specific solution required by (i), there must be a set of solutions $g_{a'b'}$ related to $g_{ab}$ by gauge transformations that

    (a)  Preserve the field equations and
    (b)  Have corresponding coordinate transformations from $x^a$ to $x^{a'}$ such that $g_{a'b'} = (\partial x^a/\partial x^{a'})(\partial x^b/\partial x^{b'})g_{ab}$.

Each of these three conditions—(i), (ii.a) and (ii.b)—is non-trivial and must be checked carefully for any proposed modification of general relativity that is not of higher-curvature, scalar-tensor or some related form. The set of solutions $g_{a'b'}$ in (ii) should be sufficiently large to include all geometries of the desired form, such as spherically symmetric metrics in models of non-rotating black holes. Condition (ii.b) then ensures that an invariant line element $\mathrm{d}s^2 = g_{ab}\mathrm{d}x^a\mathrm{d}x^b$ can be constructed from solutions of the field equations. This condition is therefore crucial, but it has often been overlooked. There are models proposed in loop quantum gravity in which line elements are used even though none of the conditions (i), (ii.a) and (ii.b) have been checked. Two such examples [23,26] are briefly discussed below. While other models propose at least some form of field equations in line with condition (i), the two remaining conditions, (ii.a) and (ii.b), have rarely been checked explicitly. For instance [8], discussed in Section 2.3.1 below, checked neither (ii.a) nor (ii.b), but it is now known that it is impossible for both conditions to be realized in such constructions [2,4]. The model proposed in [7], discussed in Section 2.3.2 below, successfully checked conditions (i) and (ii.a), but left (ii.b) open. As shown in [1], condition (ii.b) is, in fact, not met by the model of [7], but a weakened form, unnoticed in [7], can be derived in which the classical structure of space-time, implemented in (ii.b) by reference to the tensor-transformation law of Riemannian geometry, is modified. This result is an example of modified space-time structures discussed in detail in Section 3. At present, no model is known in loop quantum gravity that obeys all three conditions (i), (ii.a) and (ii.b) without a generalization from (ii.b) to

(ii.b′) The gauge transformations are such that their classical limit has corresponding coordinate transformations from $x^a$ to $x^{a'}$ with $g_{a'b'} = (\partial x^a/\partial x^{a'})(\partial x^b/\partial x^{b'})g_{ab}$.

Even though the classical theory is known to be covariant and consistent with the tensor-transformation law, experience with models of loop quantum gravity shows that embedding the condition (ii.b′) in a theory with modified gauge transformations is non-trivial because condition (ii.a) must be fulfilled before the classical limit can be taken in order for the modified theory to be well-defined.

In [23], which violates all three conditions, it is taken for granted that quantum-gravity effects can always be described by modified line elements. For instance, the proposal of an astrophysical object of Planckian density is first described as "The main hypothesis here is that a star so compressed would not satisfy the classical Einstein equations anymore, even if huge compared to the Planck scale." After further specifications, the paper continues with "Let us write a metric that could describe the resulting effective geometry." Here, the justified assumption that Einstein's equation may be modified at high density because of quantum-gravity effects is directly turned into the unsupported postulate that the corresponding geometry should be Riemannian, described by a metric tensor that determines coefficients in a line element. Similarly, Reference [26], which presents a more refined metric for bounce-based black holes, erroneously states that "the technical result of the present paper is that such a metric exists for a bouncing black to white hole" even though the paper did not actually show that a

metric of any kind exists that could be used to describe the desired effect as a solution of covariant equations. The same paper concludes with statements such as "the metric we have presented poses the problem neatly for a quantum gravity calculation. The problem now can be restricted to the calculation of a quantum transition in a finite portion of spacetime" and "this is precisely the form of the problem that is adapted for a calculation in a theory like covariant loop quantum gravity", claiming that "the spinfoam formalism is designed for this." The calculation of a quantum transition amplitude is not sufficient because it must first be shown that a quantum theory of gravity used in such a derivation does, in fact, allow a metric structure to describe its solutions. This task has not been performed in the spin-foam formalism or loop quantum gravity, in spite of the epithet "covariant" assigned to it in the preceding quote.

The tacit assumption that any solution of quantum gravity must be of metric form fails to recognize the non-trivial nature of the availability of line elements. In particular in background-independent approaches to quantum gravity, the structure of space-time is to be derived, not to be assumed. A detailed analysis should then be performed to see whether line elements are available. This conclusion refers to line elements of a generic form, setting aside the question of what their precise coefficients might be.

Spin-foam models are ill-suited for questions about space-time structure because it has not even been shown whether they are consistent space-time theories. In particular, it has not been shown that their discrete path-integral measure is covariant; see [27]. The canonical formulation of (quantum) gravity is better equipped to analyze covariance questions because it is closely related to the general consistency condition that constraints or their quantizations should be first-class and free of anomalies. Checking this condition may be complicated, but it is well-defined and amenable to systematic methods. The canonical formulation also allows one to work out effective descriptions which take into account detailed properties of quantum states [28–30]. We will first review salient features of such effective equations, and then return to the question of covariance.

*2.2. Modifications Suggested by Loop Quantum Cosmology*

In models of loop quantum gravity, modifications of the classical equations for black holes are usually based on what has been studied for some time in isotropic cosmological systems. In this context, the Friedmann equation

$$\left(\frac{\dot{a}}{a}\right)^2 = \frac{8\pi G}{3}\rho \tag{3}$$

for the scale factor $a$ is first presented in canonical form,

$$C = V\rho - 6\pi G V p_V^2 = 0 \tag{4}$$

where

$$p_V = -\frac{1}{4\pi G}\frac{\dot{a}}{a} \tag{5}$$

is canonically conjugate to the volume, $V = a^3$. The constraint $C = 0$ therefore replaces the Friedmann equation.

In loop quantum cosmology [31,32], it is argued that periodic functions of $p_V$ (or of some other combination of the canonical variables, usually linear in $p_V$ but not necessarily in $V$) should be used instead of polynomials, modeling matrix elements of holonomies for compact groups (in particular, the spatial rotation group used in loop quantum gravity). A modified constraint of the form

$$C_{\text{modified}} = V\rho - 6\pi G V \frac{\sin^2(\delta p_V)}{\delta^2} \tag{6}$$

with an ambiguity parameter $\delta$ then implies that the energy density is bounded on solutions of $C_{\text{modified}} = 0$. If this constraint is used without any further quantum corrections, Hamilton's equations generated by $C_{\text{modified}}$ can be rewritten as a modified Friedmann equation [33]

$$\left(\frac{\dot{a}}{a}\right)^2 = \frac{8\pi G}{3}\left(\rho - \frac{\rho^2}{\rho_{\text{max}}}\right) \tag{7}$$

with the maximum density

$$\rho_{\text{max}} = \frac{6\pi G}{\delta^2} \tag{8}$$

implied on solutions of the modified constraint $C_{\text{modified}} = 0$. If $\rho = \rho_{\text{max}}$, $\dot{a} = 0$ and the scale factor has a turning point.

The modified Friedmann Equation (7) indicates that bounces may be possible in models of loop quantum gravity. However, it does not present conclusive evidence because it incorporates only one out of several possible quantum effects. It does represent the characteristic loop behavior of isotropic models [34] because the classical quadratic dependence of the constraint on the momentum is replaced by a periodic function, which can be interpreted as a matrix element of a gravitational holonomy. The latter, rather than momentum components themselves, are represented as operators on the kinematical Hilbert space of loop quantum gravity [35,36]. An analogous property is encoded in the modification given by $C_{\text{modified}}$.

Because loop quantum cosmology is a quantum theory, however, one also expects that general quantum effects, such as dynamical implications of fluctuations and higher moments of a state, should be relevant, in particular at high density where the simple (7) seems to imply a bounce [37,38]. These quantum effects may be ignored in a low-curvature universe model at late times, but they are highly relevant (and suppressed in the simple (7)) close to a spacelike singularity [39,40]. A detailed analysis has also revealed previously unrecognized ambiguities in a quantum version of $C_{\text{modified}}$ because the periodic function, combined with $V$, can be implemented by several inequivalent representations of the Lie algebra $\text{sl}(2, \mathbb{R})$ [41]. It therefore remains unclear how generic bounces in loop quantum cosmology are, and by extension bounces of black-hole interiors; see [24] for a detailed analysis.

Another major problem is relevant in loop-based models of black holes. The parameter $\delta$ in $C_{\text{modified}}$, which is crucial in determining properties of bouncing solutions, is related to the characteristic scale of a specific kind of quantum effect. Its presence is motivated by the application of holonomy operators in the full theory of loop quantum gravity, in which case $\delta$ is related to the length of a curve along which parallel transport is computed. In proposed constructions of Hamiltonian constraint operators [42–48], the curve is related to links of a state acted on by the holonomy, expressed in the spin-network basis. If there is any relationship between loop quantum cosmology and loop quantum gravity, the single parameter $\delta$ therefore has to encode detailed properties of an underlying dynamical state relevant for cosmological or astrophysical evolution.

Needless to say, it is at present impossible to derive a value for $\delta$ from the full theory, but one may nevertheless use such a modification to study possible outcomes of the loop representation. However, for reliable conclusions the parameter should be put into model equations in a sufficiently general form. In particular, the value of $\delta$ may have to be adjusted, or renormalized, as the underlying space-time state evolves. Instead of a constant $\delta$, one should therefore use a function that depends on a relevant scale, such as the energy density or curvature in a certain range of evolution. As the scale changes, $\delta$ does too, which may be modeled as a certain function of $a$, a simpler parameter related to the energy scale on solutions of the Friedmann equation. Again, it is at present impossible to derive a specific function $\delta(a)$ from a space-time state, and therefore a sufficiently general ansatz is required for generic conclusions.

The large freedom in choosing such a function, compared with a single constant, implies that it is impossible to justify claims about detailed long-term effects in loop models of bounce-based

black holes. For instance, one of the claims made in [8] states that "if the radius of the black hole horizon in asymptotic region I [before the bounce] is, say, $r_B = 3$ km, corresponding to a solar mass, that of the white hole horizon in asymptotic region III [after the bounce] is $r_W \approx (3 + \mathcal{O}(10^{-25}))$ km." Unfortunately, this statement is derived from equations that use analogs of $\delta$ which are constant on solutions of the constraint and equations of motion and therefore ignore renormalization. The proposal of [8] simply assumes that a single effective theory with constant parameters can be used over a vast range of scales, stretching from the low-curvature near-horizon region of a 3 km black hole all the way to Planckian curvature at the putative bounce, and back to low curvature at a white-hole horizon of similar size. The statement not only ignores the possibility (or necessity) of renormalization, but also expects that maintaining a relative precision of $10^{-25}$ is believable over such a vast range of scales, including quantum-gravity regimes.

These problems make it difficult to draw justified conclusions about bounce-based black holes in models of loop quantum gravity. However, such models are still useful because they have revealed, somewhat unintentionally, that the effects taken from loop quantum gravity require non-trivial space-time structures in order to be meaningful. These are local space-time structures, related to the form of general covariance realized in the models. Because they are local, they are not sensitive to questions of long-term quantum evolution or renormalization, and they can be parameterized in sufficiently general form to include some quantization ambiguities. On occasion, they also rule out certain ambiguities by the condition of covariance. We will now turn to these fundamental questions, continuing in this section with canonical models that have been proposed for bounce-based black holes.

### 2.3. Violations of General Covariance

As a canonical version of the conditions given in Section 2.1, the task is to find consistent versions of the constraints of general relativity, modified such that they can incorporate bounce effects and at the same time respect general covariance in the canonical form of hypersurface deformations. We will now illustrate the highly non-trivial nature of the combination of these conditions.

#### 2.3.1. Slicing Dependence

Modifications similar to those in $C_{\mathrm{modified}}$ can be implemented in anisotropic models [49,50], including Kantowski–Sachs space-times which may describe the interior of Schwarzschild-type black holes where the timelike Killing vector field of the static exterior is replaced by a spacelike field within the horizon, implying homogeneity. On this basis, a long-standing suggestion is that collapsing space-time in a black hole could bounce back at high density, forming a non-singular black hole [25,51–56] just as a cosmological model might bounce at the big bang. However, homogeneous minisuperspace models are unable to reveal the structure of space-time that could correspond to their solutions because the only symmetries they allow are time reparameterizations. The interplay of spatial and temporal transformations, locally expressed by Lorentz boosts, remains undetermined. In particular, in minisuperspace models it is impossible to tell whether modifications such as (6) are compatible with general covariance.

An interesting suggestion to circumvent this problem has been made in [8]. As a new model of quantum-modified black holes, this proposal suffers from serious drawbacks as pointed out in [57–62] based on several independent arguments. Here, we will be interested in possible statements about space-time structure that are insensitive to properties of specific solutions. The authors point out that even the inhomogeneous exterior of a Schwarzschild black hole, by virtue of being static, allows a slicing by homogeneous hypersurfaces, but they are timelike and therefore do not correspond directly to anisotropic cosmological evolution. Nevertheless, the canonical formulation as well as quantization, or modification as in (6), can be performed on a timelike slicing. It is therefore possible to explore implications of modified dynamics in the exterior, possibly connecting it with the modified interior through a horizon in order to arrive at a complete black-hole model.

The authors of [8] worked out many details of the resulting solutions. However, they did not endeavor to determine the corresponding space-time structure, or rather to test whether there is any well-defined space-time structure at all. They instead used the same implicit assumption as in [23,26] and postulated that modifications in their anisotropic minisuperspace model can always be translated into a line element with modified coefficients. Therefore, they presupposed that their model is generally covariant without providing a proof.

As shown in [2,4], the model of [8] not only fails to be covariant, it can even be used to derive a no-go theorem for the covariance of any modification of the form (6). By taking these modifications to the exterior, the resulting equations of motion become sensitive to space-time structure for the following general reason: if a region of space-time permits a timelike homogeneous slicing, it also permits a static spherically symmetric slicing. These two slicings are related by an exchange of time and space coordinates in the models, which allows one to derive modified spherically symmetric dynamics from any proposed modified homogeneous dynamics. The anisotropic but (timelike) homogeneous slicing implies a line element of the form

$$\mathrm{d}s^2_{\mathrm{homogeneous}} = J(n)^2 \mathrm{d}n^2 - a(n)^2 \mathrm{d}t^2 + b(n)^2 \left( \mathrm{d}\vartheta^2 + \sin^2 \vartheta \mathrm{d}\varphi^2 \right) \tag{9}$$

where $n$ is a coordinate in a direction normal to timelike slices, while the time coordinate $t$ is part of the timelike homogeneous slices. The coefficient $J(n)$ is the lapse (or, rather, spatial "jump") function in a spacelike direction.

For any choice of $J$, $a$ and $b$, this line element is locally equivalent to the static spherically symmetric line element

$$\mathrm{d}s^2_{\mathrm{spherically\ symmetric}} = -a(x)^2 \mathrm{d}t^2 + J(x)^2 \mathrm{d}x^2 + b(x)^2 \left( \mathrm{d}\vartheta^2 + \sin^2 \vartheta \mathrm{d}\varphi^2 \right) \tag{10}$$

if we just rename $n$ as $x$ and use spacelike slices of constant $t$ instead of timelike slices of constant $n$. If a solution of the (modified) homogeneous model on timelike slices presents a covariant space-time, it must therefore be equivalent to a static spherically symmetric solution of some spherically symmetric model, based on the transformation of metric components implied by equating $\mathrm{d}s^2_{\mathrm{spherically\ symmetric}}$ with the generic form of a spherically symmetric line element,

$$\mathrm{d}s^2 = -N(t,x)^2 \mathrm{d}t^2 + L(t,x)^2 \left( \mathrm{d}x + M(t,x)\mathrm{d}t \right)^2 + S(t,x)^2 \left( \mathrm{d}\vartheta^2 + \sin^2 \vartheta \mathrm{d}\varphi^2 \right) \tag{11}$$

with free functions $N$, $M$, $L$ and $S$ depending only on time $t$ and the radial coordinate $x$.

Unlike homogeneous models, spherically symmetric models are strongly restricted by general covariance [63,64]: if they are local, quadratic in momenta and without higher derivatives, they must, up to field redefinitions, be of the form of $1 + 1$-dimensional dilaton gravity, with action

$$S[g,\phi] = \frac{1}{16\pi G} \int \mathrm{d}t \mathrm{d}x \sqrt{-\det g} \left( \phi R - \frac{1}{2} g^{ab} \frac{\partial \phi}{\partial x^a} \frac{\partial \phi}{\partial x^b} - V(\phi) \right) , \tag{12}$$

in which only a single function, the dilaton potential $V(\phi)$, can be changed in order to adjust the equations to potential modifications. If the condition of quadratic dependence on momenta is dropped, the form of local generally covariant actions with second-order field equations is still quite restricted, given by generalized dilaton models [65] of the form

$$
\begin{aligned}
S[g,\phi] = {} & \frac{1}{16\pi G} \int \mathrm{d}t \mathrm{d}x \sqrt{-\det g} \big( \xi(\phi) R + k(\phi, X) \\
& + C(\phi, X) \nabla^a \phi \nabla^b \phi \nabla_a \nabla_b \phi \big) .
\end{aligned}
\tag{13}
$$

Instead of a single dilaton potential, there are now three free functions, $\xi(\phi)$, $k(\phi, X)$ and $C(\phi, X)$, but crucially they can depend only on the scalar field $\phi$ and the first-order derivative expression

$$X = -\frac{1}{2} g^{ab} \nabla_a \phi \nabla_b \phi \,. \tag{14}$$

According to [66], the actions (13) present the most general 2-dimensional local scalar-tensor theories with second-order field equations, or a 2-dimensional version of Horndeski theories [67].

The canonical structure of modifications such as (6), without independent momenta not seen in the classical theory, characterizes loop-modified models as local ones without higher derivatives. The modified timelike homogeneous dynamics they imply must therefore be consistent with some choice of generalized dilaton model with static solutions, if they have a chance of being covariant. Importantly, the free functions in (13) can depend only on one of the degrees of freedom, $\phi$, and its first-order derivatives but not on the 2-dimensional metric $g_{ab}$. In a spherically symmetric interpretation, $\phi$ is determined by the function $S(t, x)$ in (11).

A detailed analysis using the canonical equations of dilaton gravity shows that it is impossible to express loop modifications in generalized dilaton form [2,4]. As a brief argument, holonomy modifications of a model with line element (9) imply a Hamiltonian that is non-polynomial in the momenta of $a$ and $b$, and therefore non-polynomial in normal derivatives of these coefficients by $n$. In the spherically symmetric interpretation, normal derivatives of $a$ and $b$ are translated into spatial derivatives of the lapse function $N$ and $S$ by $x$ in (11). Non-polynomial corrections in $\partial N / \partial x$ cannot be expressed in terms of the free functions of a generalized dilaton model. Loop modifications of the timelike homogeneous model therefore cannot be consistent with slicing independence.

The modified timelike-homogeneous dynamics in a classical space-time therefore cannot be interpreted geometrically in terms of a metric or a line element. The assessment, given in [68], that the authors of [8] "have shown that loop quantum gravity—tentative theory of quantum gravity—predicts that spacetime continues across the center of the hole into a new region that has the geometry of the interior of a white hole, and is located in the future of the black hole" is therefore incorrect because in this model there is, in fact, no space-time that could describe the modified solutions even at low curvature, let alone continue space-time across the center of the black hole. It is not true that the authors "have shown that a crucial ingredient of this scenario, the transition at the center, follows from a genuine quantum gravity theory, namely loop theory" or that "loop gravity predicts that the interior of a black hole continues into a white hole" [68].

Modifications of loop quantum cosmology are inconsistent with slicing independence, which is a consequence of classical general covariance. In the next section we will discuss a way to evade the conclusion that models of loop quantum gravity violate covariance, based on the possibility that classical symmetries may be deformed—that is, affected by quantum corrections — even if they are not violated. Such a theory, using the generalization (ii.b') of the classical condition (ii.b) in Section 2.1, would still be consistent and free of anomalies, but it would not permit an interpretation as a geometric theory with solutions based on Riemannian geometry. Before we discuss the underlying models, it is useful to consider another proposal that aimed (but ultimately failed) to evade strong restrictions from general covariance on modifications suggested by loop quantum gravity.

2.3.2. Spherically Symmetric Models

Before we are ready to discuss deformed covariance and space-time structures, we should comment on several attempts to derive black-hole models by evading the covariance question. Some of them simply ignore general covariance by fixing the space-time gauge before implementing quantization or modification, and then working only in this one gauge fixing [69–71]. These attempts need not be discussed in any detail because modifying the equations of a gauge theory after fixing the gauge leads to questionable physics unless it can be shown that a covariant quantization of this kind

exists. Without an explicit demonstration of gauge invariance, none of the conditions (i), (ii.a) and (ii.b) or (ii.b′) of Section 2.1 are guaranteed to hold.

A more refined suggestion to implement modifications similar to (6) has been made in [7], using spherically symmetric models, classically described by space-times with line elements (11). The original constructions of [7] were made in triad variables, but they are independent of this choice and hold equally in metric variables as used here. The isotropic Friedmann constraint is then replaced by the functional Hamiltonian constraint

$$H[N] = \int N \left( -\frac{p_L p_S}{S} + \frac{L p_L^2}{2S^2} + \frac{1}{2}\frac{(S')^2}{L} + \frac{SS''}{L} - \frac{SS'L'}{L^2} + \frac{1}{4}LSV(S) \right) \mathrm{d}x \qquad (15)$$

where $p_S$ and $p_L$ are canonically conjugate to $S$ and $L$, respectively. Here, for the sake of generality, we have included the dilaton potential $V(S)$ of $1+1$-dimensional dilaton gravity, which equals $V(S) = -2/S$ in spherically symmetric models reduced from $3+1$-dimensional general relativity. In addition, we have the diffeomorphism constraint

$$D[M] = \int M \left( S' p_S - L p_L' \right), \qquad (16)$$

such that

$$\{H[N_1], H[N_2]\} = D[L^{-2}(N_1 N_2' - N_1' N_2)]. \qquad (17)$$

The last condition ensures that gauge transformations generated by the Hamiltonian and diffeomorphism constraints correspond to hypersurface deformations in some classical space-time with Riemannian geometry. In any spherically symmetric theory with (17), condition (ii.b) of Section 2.1 holds.

Modifying the Hamiltonian constraint of spherically symmetric models is a much more non-trivial exercise than modifying the single constraint of isotropic cosmological models. Most attempts simply break the underlying classical symmetries, such that the Poisson bracket $\{H[N_1], H[N_2]\}$ is not related to any of the relevant generators, $H[N]$ and $D[M]$. An interesting observation was made in [7] which makes it easier to incorporate modified constraints: the linear combination

$$H[2PS'/L] + D[2Pp_L/(SL)] = \int P \frac{\mathrm{d}}{\mathrm{d}x} \left( -\frac{p_L^2}{S} + \frac{S(S')^2}{L^2} + \frac{1}{2} \int SV(S)\mathrm{d}S \right) \mathrm{d}x \qquad (18)$$

of constraints does not depend on $p_S$, and the remaining terms form a complete spatial derivative, except for the new multiplier $P$. As a constraint, the linear combination can therefore be replaced by a new constraint,

$$C[Q] = \int Q \left( -\frac{p_L^2}{S} + \frac{S(S')^2}{L^2} + \frac{1}{2} \int SV(S)\mathrm{d}S + C_0 \right) \mathrm{d}x, \qquad (19)$$

with a free constant $C_0$, and $Q$ is now a multiplier with density weigtht minus one.

The density weight implies that the bracket of $C[Q]$ with the diffeomorphism constraint is

$$\{C[Q], D[M]\} = -C[(MQ)'], \qquad (20)$$

while it is easy to see that

$$\{C[Q_1], C[Q_2]\} = 0 \qquad (21)$$

because $C[Q]$ depends only on the momentum $p_L$ and on none of the spatial derivatives of $L$. The Poisson bracket therefore produces only delta functions but none of their spatial derivatives, and all terms cancel out in the antisymmetric $\{C[Q_1], C[Q_2]\}$. A non-Abelian constraint $H[N]$ with

structure functions in the bracket (17) has therefore been replaced by a partially Abelian constraint $C[Q]$ with structure constants.

The general properties of (19) that imply (21) remain unchanged if $p_L^2$ in (19) is replaced with an arbitrary function of $p_L$. The modified constraint then still depends only on the momentum $p_L$, and not on spatial derivatives of $L$. Moreover, since $p_L$ has density weight zero, the bracket (20) is not affected by the modification. It therefore seems possible to implement an arbitrary $p_L$-dependent modification of (19) without changing the brackets of constraints, just as it is possible to modify (6) in isotropic models. In particular, the modified constraints remain first class (they obey condition (ii.a) of Section 2.1) and could be expected to describe a generally covariant theory that could be expressed in terms of modified line elements.

However, this conclusion ignores the non-trivial nature of hypersurface-deformation brackets, or of condition (ii.b) [1]. For a covariant theory, it is not sufficient to have an anomaly-free system of constraints with closed brackets such as (20) and (21). The generators should also correspond to hypersurface deformations in space-time. Since the partially Abelianized brackets (20) and (21) are not of hypersurface-deformation form, for a covariant space-time theory we must be able to form linear combinations of the constraints such that a bracket of the form (17) is obtained, providing the correct relationship required for normal deformations of hypersurfaces in space-time [72–74]. Without this demonstration, the model obeys condition (ii.a) but not condition (ii.b) from Section 2.1, and therefore could not be used to derive effective line elements.

After modifying the Abelianized constraint $C$, as in

$$C[Q] = \int Q \left( -\frac{f_1(p_L)}{S} + \frac{S(S')^2}{L^2} + \frac{1}{2} \int SV(S)\mathrm{d}S + C_0 \right) \mathrm{d}x \tag{22}$$

with some modification function $f_1$, we must therefore be able to retrace our steps that led to the definition of $C$ if the modification is to preserve general covariance. Given the form of the diffeomorphism constraint and the known expression of the classical Hamiltonian constraint, which should be obtained in the "classical" limit where $f_1(p_L) \to p_L^2$, the only combination that could equal the modified $C[Q]$, or the derivative expression (18) from which it is derived, is

$$
\begin{aligned}
&\int P \frac{\mathrm{d}}{\mathrm{d}x} \left( -\frac{f_1(p_L)}{S} + \frac{S(S')^2}{L^2} + \frac{1}{2} \int SV(S)\mathrm{d}S \right) \mathrm{d}x \\
=\ & \int 2P \left( -\frac{p_L'}{2S}\frac{\mathrm{d}f_1}{\mathrm{d}p_L} + \frac{S'}{L}\left( \frac{Lf_1(p_L)}{2S^2} + \frac{(S')^2}{2L} + \frac{SS''}{L} - \frac{SS'L'}{L^2} + \frac{LSV(S)}{4} \right) \right) \mathrm{d}x \\
=\ & \tilde{H}[2PS'/L] + D[2Pf_2(p_L)/(SL)]
\end{aligned}
\tag{23}
$$

with some function $f_2(p_L)$ in the multiplier of the diffeomorphism constraint, and a possibly modified Hamiltonian constraint $\bar{H}[N]$. This equation, together with an unmodified diffeomorphism constraint because the canonical formulation assumes the classical structure of space at equal times, implies that

$$f_2(p_L) = \frac{1}{2}\frac{\mathrm{d}f_1}{\mathrm{d}p_L}. \tag{24}$$

This function is uniquely determined by the single term in (23) that depends on $p_L'$. Subtracting $D[2Pf_2(p_L)/(SL)]$ from (23) then implies

$$\tilde{H}[N] = \int N \left( -\frac{f_2(p_L)p_S}{S} + \frac{Lf_1(p_L)}{2S^2} + \frac{1}{2}\frac{(S')^2}{L} + \frac{SS''}{L} - \frac{SS'L'}{L^2} + \frac{1}{4}LSV(S) \right) \mathrm{d}x. \tag{25}$$

This modified constraint is free of anomalies for any function $f_1(p_L)$, such that $f_2$ is determined by (24), because it has just been derived as a linear combination of constraints in an anomaly-free, partially Abelianized system. However, the restrictive nature of general covariance in the

form of hypersurface-deformation generators, canonically encoding condition (ii.b) of Section 2.1, can nevertheless be seen. For instance, if $f_1$ depends on $S$ in addition to $p_L$, as suggested in [75], such as $S^{-2j} f_1(p_L S^j)$ with some constant $j$ which may be subject to renormalization, the combination used in (23) includes a term $-\frac{1}{2} j S^{-j-2} S' p_L (\mathrm{d} f_1/\mathrm{d} z)|_{z=p_L S^j}$. Hypersurface-deformation generators can still be reconstructed from a modified (22) with a similar relationship, $f_2(p_L S^j) = \frac{1}{2} S^{-j} (\mathrm{d} f_1/\mathrm{d} z)|_{z=p_L S^j}$, between the two modification functions in (22) and in the multiplier of the diffeomorphism constraint, respectively.

However, the reconstructed modified Hamiltonian constraint is of the form (25) with $f_1$ replaced by

$$\bar{f}_1(p_L S^j) = (2j+1) \frac{f_1(p_L S^j)}{S^{2j}} - 2j p_L f_2(p_L S^j). \tag{26}$$

While the resulting modified system is anomaly-free, it is not possible for all three functions in (26), $f_1$, $f_2$ and $\bar{f}_1$, to be periodic in their argument. It is therefore more difficult to motivate these modifications by holonomy terms. This subtlety had already been pointed out in [12,76], but it went unnoticed in [75] because general covariance was not analyzed in this paper (while effective line elements were nevertheless proposed). In certain other systems that may be partially Abelianized, such as spherically symmetric gravity with a scalar field [77], one can show that no hypersurface-deformation generators exist [1]. Even though anomaly-free quantizations or modifications can then be found in the partially Abelian system, they cannot be considered covariant.

In the vacuum case, while the modified constraint (25) is anomaly-free, it obeys a bracket

$$\{\tilde{H}[N_1], \tilde{H}[N_2]\} = D[L^{-2} \beta(p_L)(N_1 N_2' - N_1' N_2)] \tag{27}$$

where

$$\beta(p_L) = \frac{1}{2} \frac{\mathrm{d}^2 f_1}{\mathrm{d} p_L^2} \tag{28}$$

if $f_1$ depends only on $p_L$ (and a related expression if $f_1$ depends on both $p_L$ and $S$ through the combination $p_L S^j$; see [10]). If $\beta(p_L) \neq 1$, the symmetries are modified compared with the classical relationship (17). They no longer correspond to hypersurface deformations in space-time, and are not dual to coordinate transformations. The model of [7] is not consistent with condition (ii.b) of Section 2.1, but it is compatible with the generalized form (ii.b'). Modifying partially Abelianized constraints (19) therefore produces modified space-time structures in disguise. It is then not guaranteed that line elements obtained by inserting solutions of modified constraints are meaningful because such solutions would be subject to gauge transformations that are not dual to coordinate changes of $\mathrm{d} x^a$. A detailed analysis of modified space-time structures is therefore required.

## 3. Modified Space-Time Structure

Well before [7] was published, modified brackets of the form (27) had already been found by analyzing space-time structure directly in terms of generators of hypersurface deformations [10–13]. These studies followed the usual reasoning of effective field theories, in which potential quantum effects are explored in a theory of classical type by including quantum modifications as well as other terms of the same order that are consistent with all required symmetries, here given by general covariance as represented by hypersurface deformations. For a detailed discussion of effective field theory applied to the canonical formulation of spherically symmetric models, see [78].

### 3.1. Anomaly-Freedom

In a Lagrangian treatment, an effective derivation of the covariance condition usually leads to higher-curvature effective actions, perhaps with additional independent degrees of freedom in terms of new fields. Symmetries that determine the structure of allowed contributions to an effective action up to a given order in derivatives can be evaluated systematically by using the tensor-transformation

law. Higher-derivative terms must then be curvature invariants, or suitable combinations of derivatives of new fields.

In a Hamiltonian formulation, considerations of the space-time tensor transformation law cannot be used directly because space-time fields are decomposed into components according to a foliation of space-time. Tensor transformations are replaced by closure conditions imposed on the Poisson brackets of the Hamiltonian and diffeomorphism constraint, which directly refer to the central objects of a Hamiltonian formulation. A canonical theory is generally covariant provided it has constraints that generate hypersurface deformations, such that (17) is fulfilled [74]. Higher-curvature effective actions indeed obey this relationship, irrespective of their coefficients of higher-curvature terms [79].

The Hamiltonian treatment is technically more involved than the Lagrangian one, but it is important because it also allows for generalizations of covariance: the closure condition is more general than an application of the tensor transformation law because the latter, but not the former, presupposes that space-time is equipped with Riemannian geometry. In the language of hypersurface deformations, Riemannian geometry is equivalent to the choice of $\beta = 1$ in (27), or $\beta = -1$ in Euclidean signature. The existence of consistent spherically symmetric models with $\beta \neq \pm 1$ proves that the Hamiltonian treatment is more general than Lagrangian considerations based on the tensor-transformation law of Riemannian geometry.

When one tries to implement holonomy modifications in spherically symmetric models using the language of effective field theory, one can start with the expression (25) already encountered in the preceding section. The two functions $f_1$ and $f_2$ are initially arbitrary and allow for a modified, non-quadratic dependence on the momentum $p_L$. While the dependence on $p_S$ could also be expected to be modified, no covariant version of this form has been found yet. A possible explanation [12] is that $p_S$, unlike $p_L$, has a spatial density weight in spherically symmetric space-times, and therefore needs to be integrated spatially before it can be inserted in a non-linear function with well-defined spatial transformation properties. Such modifications would therefore be non-local or, in a derivative expansion, give rise to higher spatial derivatives. For such terms to be consistent in an effective treatment, one would also have to include a series of higher-derivative modifications of the spatial derivatives of $L$ and $S$ already present in (25), considerably complicating derivations of the Poisson brackets of constraints. It is therefore more difficult to find modifications of the $p_S$-dependence, but such a modification has not been ruled out yet. Similar considerations apply to studies that aim to modify the full theory because all components of the gravitational momentum then have non-scalar transformation properties with respect to spatial coordinate changes. Simple modifications that do not include higher spatial derivatives have indeed been ruled out [80].

Two Hamiltonian constraints of the form (25) have a Poisson bracket

$$\{\bar{H}[N_1], \bar{H}[N_2]\} = \int (N_1 N_2' - N_1' N_2) \left( \frac{S'}{LS} \left( f_2(p_L) - \frac{1}{2} \frac{df_1}{dp_L} \right) + \frac{1}{L^2} \frac{df_2}{dp_L} (p_S S' - L p_L') \right) dx. \quad (29)$$

The first term on the right-hand side does not vanish on the constraint surface of Hamiltonian and diffeomorphism constraints. It must therefore vanish for an anomaly-free implementation of modifications, which requires [10]

$$f_2(p_L) = \frac{1}{2} \frac{df_1}{dp_L}. \quad (30)$$

This is the same condition (24) found in the preceding section, but derived independently.

The Poisson bracket of two Hamiltonian constraints is then proportional to the diffeomorphism constraint, and there are no anomalies. It is of the form (27) with

$$\beta(p_L) = \frac{df_2}{dp_L} = \frac{1}{2} \frac{d^2 f_1}{dp_L^2}, \quad (31)$$

a function that, in general, is not equal to $\pm 1$. The consistent modifications in (25) with (30) therefore imply a non-Riemannian space-time structure. For small $p_L$, $\beta \to 1$ provided $f_1(p_L) \to p_L^2$ to respect the classical limit. If $f_1$ is a bounded function with a local maximum, we have $\beta < 0$ around the local maximum, which corresponds to a non-classical version of space-time with Euclidean signature. For instance, if $f_1(p_L) = \delta^{-2} \sin^2(\delta p_L)$ as in the isotropic $C_{\text{modified}}$, we have

$$\beta(p_L) = \cos(2\delta p_L) \approx -1 \tag{32}$$

around local maxima of $f_1$, where $\delta p_L = (k + 1/2)\pi$ with integer $k$.

The modified Hamiltonian constraint (25) together with the usual diffeomorphism constraint (16) define a canonical system that is free of anomalies and covariant in a generalized sense [16], respecting condition (ii.b') of Section 2.1: the bracket (27), together with the unmodified classical bracket between $\bar{H}[N]$ and $D[M]$, shows that the set of constraints is not only closed algebraically, but also such that hypersurface deformations are obtained in the classical limit, $\beta \to 1$. The second property defines generalized covariance, and it is not implied by the first property, the closure condition.

### 3.2. Signature Change

For $\beta \neq 1$ in (27), algebraic relations between the constraints are modified compared with the classical form of hypersurface deformations. Therefore, the transformations they generate cannot be interpreted as changes of tensor fields on a Riemannian space-time canonically foliated by hypersurfaces. In this way, generalized covariance is able to evade the no-go results discussed in Section 2.3.1 because the latter assume the classical form of general covariance which is equivalent to slicing independence in space-time. At present, it is not known how to interpret the transformations of generalized covariance in geometrical language under all circumstances, but for solutions such that $\beta$, through $p_L$, depends only on time it has been shown in [81] that a field redefinition of metric components can be used to map the space-time description to Riemannian form. This case may be used in the interior region of a Schwarzschild black hole within the horizon, using a gauge that implies homogeneous spatial slices. It is therefore instructive in the present context, even though the inhomogeneous exterior where $\beta$ depends on the spatial position appears to require a more involved geometrical description in which, according to the no-go results of Section 2.3.1, slicing independence cannot easily be seen.

In the interior, the field redefinition derived in [81] leads to an effective line element that is compatible with modified gauge transformations generated by (25). The space-time line element requires transformations of all metric components, not just of the spatial part $q_{ab}$ which provides phase-space degrees of freedom and can directly be transformed by computing the Poisson bracket

$$\{q_{ab}, H[\epsilon^0] + D[\epsilon^i]\} = \mathcal{L}_\xi q_{ab} \tag{33}$$

where $\epsilon^0$ and $\epsilon^i$ are such that the space-time vector field $\xi^a$ that appears in the Lie derivative $\mathcal{L}_\xi$ has components

$$\xi^0 = \frac{\epsilon^0}{N} \quad , \quad \xi^i = \epsilon^i - \frac{N^i}{N}\epsilon^0 . \tag{34}$$

These components rewrite the vector field $(\epsilon^0, \epsilon^i)$ expressed in a basis adapted to the foliation by using the unit normal $n^a$ into components with respect to a coordinate basis with the usual ADM-like time direction $t^a = Nn^a + N^a$ [73] with the lapse function $N$ and spatial shift vector $N^a$ on a given space-time on which the transformation is performed [82].

The remaining components of the space-time metric, given by lapse and shift in

$$ds^2 = -N^2 dt^2 + q_{ij}\left(dx^i + N^i dt\right)\left(dx^j + N^j dt\right) , \tag{35}$$

are subject to gauge transformations derived from the condition that (33) is consistent with the time direction determined by these functions. Based on this condition, lapse and shift transform according to [83]

$$\delta_\epsilon N^a(x) = \dot{\epsilon}^a(x) + \int d^3y d^3z N^b(y) \epsilon^c(z) F^a_{bc}(x,y,z) \tag{36}$$

where $F^a_{bc}(x,y,z)$ are (distributional) structure functions of the constraints, $C_a[\Lambda^a] = H[\Lambda^0] + D[\Lambda^i]$, such that

$$\{C_a[\Lambda^a_1], C_b[\Lambda^b_2]\} = C_c[\Lambda^c_{12}] \tag{37}$$

with

$$\Lambda^c_{12}(x) = \int d^3y d^3z F^c_{ab}(x,y,z) \Lambda^a_1(y) \Lambda^c_2(z). \tag{38}$$

A modified bracket (27) implies a non-classical $F^i_{00}$, and therefore changes the gauge transformation of the shift components, in addition to modified gauge transformations implied by (33) with a modified (25). As shown by detailed derivations in [81] for the case of a spatially constant $\beta$ in spherical symmetric space-times, the new gauge transformations are dual to coordinate changes of $dx^a$, such that a suitable effective line element $ds^2 = \bar{g}_{ab} dx^a dx^b$ is invariant. The components of the effective metric $\bar{g}_{ab}$ are of the form (11), except that $N^2$ is replaced with $\beta N^2$ [81]:

$$ds^2 = -\beta(t) N(t,x)^2 dt^2 + L(t,x)^2 (dx + M(t,x)dt)^2 + S(t,x)^2 \left(d\vartheta^2 + \sin^2 \vartheta d\varphi^2\right). \tag{39}$$

An effective line element therefore exists in this case which expresses generalized covariance in terms of Riemannian geometry after a field redefinition.

This line element shows directly that $\beta < 0$ implies a transition to Euclidean signature. The line element is degenerate when $\beta = 0$ and therefore does not describe a valid Riemannian geometry at the transition hypersurface between Lorentzian and Euclidean signature. It therefore provides two distinct effective descriptions of a single canonical solution, corresponding to the regions where $\beta > 0$ and $\beta < 0$, respectively. These two regions can be bridged in the canonical theory, whose equations remain valid at $\beta = 0$ [80,81,84], but there is no Riemannian interpretation of the transition surface. Since the two regions are separated by a hypersurface of codimension one, it is possible to extend fields across the transition surface by taking limits in one region approaching the surface, and using the limiting values as initial conditions for an extension into the other region. However, the existence of such a mathematical extension does not necessarily imply a causal relationship.

*3.3. Signature Change and Non-Singular Space-Time*

Fields on a space-time with line element (39) obey partial differential equations of mixed type, which are hyperbolic in the Lorentzian region and elliptic in the Euclidean region [85]. A well-posed problem for solutions therefore requires a mixture of initial values in the Lorentzian region and boundary values in the Euclidean region, which latter appear as final conditions in a temporal interpretation based on the Lorentzian phase; see Figure 1. The transition surface therefore cannot be bridged by deterministic evolution, even though continuous and well-behaved mathematical extensions are possible for given final conditions. The precise form of relevant initial-boundary value problems has been specified by Tricomi [86].

It has occasionally been claimed that signature change in models of loop quantum cosmology has been ruled out by observations, but such arguments are based on a single discredited study [87], which erroneously used a standard initial-value problem throughout the high-density phase. It is not necessary to compute a power spectrum, as done in [87], in order to rule out an ill-posed problem because of implied instabilities. Discrepancies in the power spectrum derived in [87] compared with observations therefore do not rule out signature change; they are only a symptom of the incorrect treatment of initial values. A clarification of this problem has been published by some of the authors in [88]. The relevant issues had already been described in [85]. In this context, it is important to note

that signature change is ubiquitous in models of loop quantum gravity. It appears not only in a direct treatment of modified constraints of the form (25), but is also realized in a hidden way in partially Abelianized treatments based on (22), as described in Section 2.3.2.

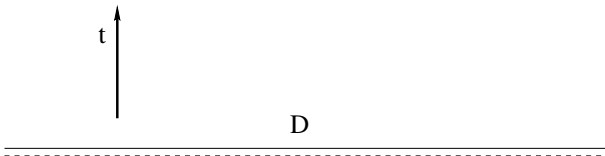

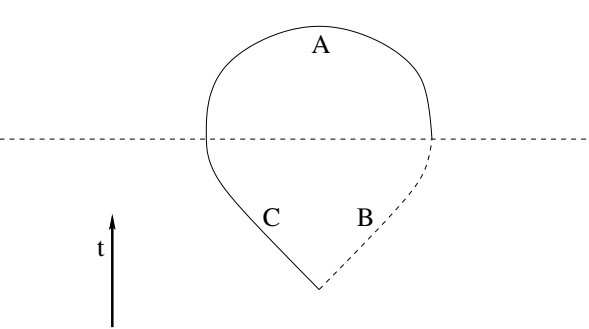

**Figure 1.** Well-posed initial-boundary value problem for mixed-type partial differential equations in two dimensions, according to Tricomi [86]. Arrows indicate the assumed flow of time in Lorentzian regions, while the two horizontal dashed lines are transition surfaces between two Lorentzian regions (top and bottom) and a Euclidean central region, as it may be realized at the center of a bouncing cosmological or black-hole solution. Entering the Euclidean region (bottom), the Tricomi problem shows that smooth data for the field (but not its normal derivatives) posed on the union of one characteristic in the original Lorentzian region (C) as well as an arc (A) in the Euclidean region connecting the end point of C with the end point of the other characteristic (B) starting at the same point as C imply a unique smooth solution in the region bounded by A, B and C, which depends continuously on the data. The solution cannot be freely specified on B. Data on A amount to final values as seen from the flow of time in the initial Lorentzian region. On the other side of the Euclidean region, top, the flow of time points away from the transition surface. A standard initial-value problem can therefore be used on any equal-time hypersurface (D) after the transition ("after" defined according to the flow of time in the Lorentzian region pointing away from the Euclidean region). In this initial-value problem, both the field and its normal (that is, time) derivative can be chosen freely on D.

Signature change reveals a new possibility to avoid singularity theorems of general relativity. Unlike a simple loop-motivated bounce based on (6), it clearly shows which of the usual assumptions of these theorems no longer hold. The main theorems are insensitive to the actual dynamics and only use energy conditions as well as properties of Riemannian geometry such as the geodesic deviation equation, in addition to topological assumptions. It is then difficult to understand how vacuum solutions for black holes can be singularity free, as in bouncing proposals, based on modified dynamics while maintaining Riemannian geometry and (because there is no matter) energy conditions. Of course, it is always possible to write corrections to Einstein's equation as an effective stress-energy tensor, which may formally violate energy conditions, but such a reformulation is not necessary and therefore does not show how singularity theorems are evaded. Moreover, rewriting a modification of Einstein's equation through an effective stress-energy tensor assumes that the theory is covariant and compatible with Riemannian geometry, which is not guaranteed.

While scenarios of dynamical signature change are non-singular, unlike classical signature change [89–93], they do not violate singularity theorems. Modified hypersurface deformations

according to (27) imply that the entire space-time structure, including both Lorentzian and Euclidean regions, is not Riemannian. For this reason, the usual theorems do not apply. Riemannian geometry can be used after a field redefinition that leads to the effective line element (39), but only in two disjoint regions of Lorentzian and Euclidean signature, respectively. If we start in the effective Lorentzian region, timelike geodesics are indeed inextendible, as required by singularity theorems, because the transition surface of signature change is reached after a finite amount of proper time. Thereafter, time, and therefore timelike geodesics, do not exist. However, the transition surface is not a boundary of space(-time) in the generalized sense that allows for signature change. Geodesics can be extended across this surface as spacelike ones, as illustrated in Figure 2. Such an extension requires final values in the Euclidean region, just as shown by the Tricomi problem for well-posed problems of mixed-type partial differential equations.

### 3.4. Evaporation Scenarios Ruled Out by Signature Change

Based on loop quantum gravity, bounce-based black holes as suggested in [23] are ruled out by signature change. While there may be no singularity in a space-time modified by effects from loop quantum gravity, the structure of space-time is modified such that deterministic evolution through high curvature is impossible. The putative bounce as a dynamical process is stopped in its tracks by signature change and does not happen. Matter that collapsed or fell into the black hole does not reappear later because it cannot evolve through timeless high density or curvature. While a white hole might open up in the future where the black hole had been, the necessity of final conditions, according to Figsure 1 or Figure 2, implies that it is not uniquely determined by the previous state through the complete lockdown imposed by the Euclidean phase. It therefore represents a naked singularity: specifying evolution to the future of the Euclidean region requires a choice of new initial values on the top line (D) in Figure 1 which are not determined by the state of the initial black hole; see Figure 3 for a possible embedding of the Euclidean region in a space-time diagram.

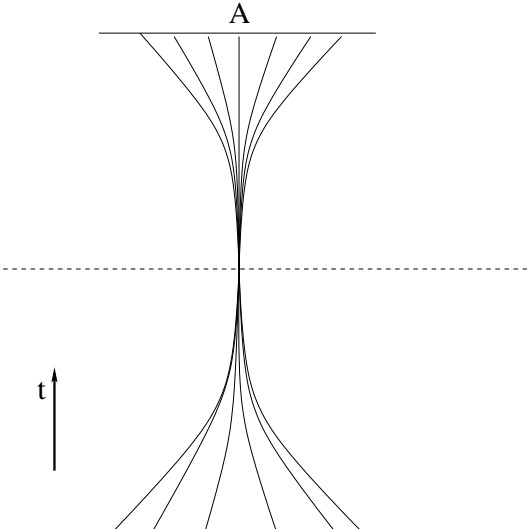

**Figure 2.** Final conditions required for an extension of geodesics through a hypersurface of signature change (dashed). A $\beta$ approaching zero in the effective line element (39) implies that light cones in the Lorentzian region collapse. All timelike geodesics aimed toward a given point on the transition surface therefore arrive there with the same asymptotic direction. Using the limiting values as initial values for extended (now spacelike) geodesics in the Euclidean region, a unique extension follows only if additional data are provided, such as the final point of a spacelike geodesic. The final points for a family of geodesics, lined up in this figure along a curve A, correspond to the final data on the arc A in the Tricomi problem, Figure 1.

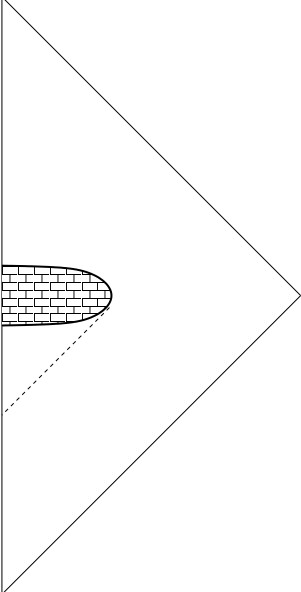

**Figure 3.** A bouncing interior of a Planck star is ruled out by signature change because the high-density region (indicated as a brick wall) is Euclidean and does not allow deterministic evolution. Boundary values around the Euclidean region, required for a well-posed problem of Tricomi-type, imply indeterministic behavior as shown in Figure 4.

More generally, any black-hole interior reopening after the Euclidean lockdown and connecting with the original exterior is ruled out because it would violate deterministic behavior at low curvature: an exterior observer who always stays at low curvature would suddenly be inundated by whatever data may have been posed on the future boundary of the Euclidean region, unbeknownst to the low-curvature observer. The rightmost edge of the Euclidean region is the starting point of a Cauchy horizon [94] because it marks the transition into a region no longer determined by the distant past; see Figure 4. Even though curvature remains finite, the top boundary of the Euclidean region presents a naked singularity.

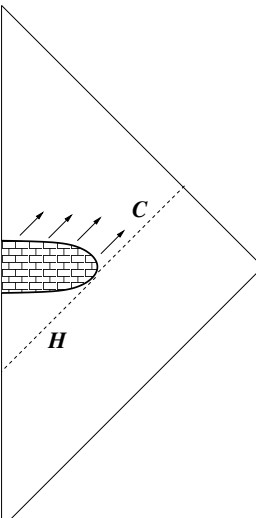

**Figure 4.** A Euclidean core implies a Cauchy horizon *C* in addition to the event horizon *H*, if it opens up in the original exterior. Observers crossing *C* at low curvature will be exposed to undetermined information from the future boundary of the Euclidean region where new initial values must be posed for the future half of this Penrose diagram.

*3.5. Evaporation Scenarios Consistent with Signature Change*

Signature change does not always imply unacceptable violations of deterministic behavior. There is certainly no determinism in a Euclidean region because there is no time, but as long as this region is confined to high curvature and does not have implications on observers who always stay at low curvature, this behavior is not ruled out by common requirements on fundamental physics.

A possible consistent scenario is given by a model in which the initial black-hole interior does not open back up into the original exterior after the Euclidean region. Instead, it forms an instantly orphaned baby universe that lacks complete causal contact with a parent; see Figure 5. While data on the future edge of the Euclidean region remain undetermined by the black-hole past and technically play the role of a naked singularity, this is no different from the initial singularity we are used to from classical cosmological models. The naked singularity sets the stage for a new universe, but, unlike in Figure 4, it does not affect observers who always stayed at low curvature. This scenario is therefore permissible.

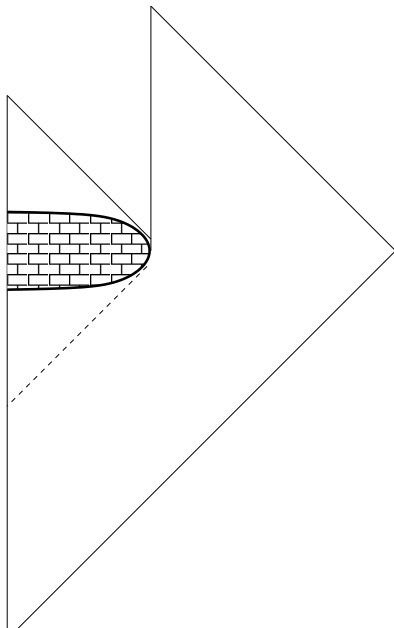

**Figure 5.** An orphan universe starting at the future edge of the Euclidean core without connecting to the former black-hole exterior.

It is not necessary to include the orphan universe in the space-time diagram. Instead, the Euclidean region could be a final boundary of the black-hole interior on which future data are posed but not evolved further; see Figure 6. In this form, no naked singularity appears, or no singularity at all because the Euclidean region which eliminates the usual curvature singularity of Schwarzschild black holes now lacks a future Lorentzian region and does not initiate a new space-time region unrelated to its past. This consistent scenario is closely related to the independent proposal of [95], also explored in a related way in [96].

While the scenarios of an orphan universe and a final state are consistent with commonly accepted deterministic behavior, they do not solve the information loss problem of black holes.

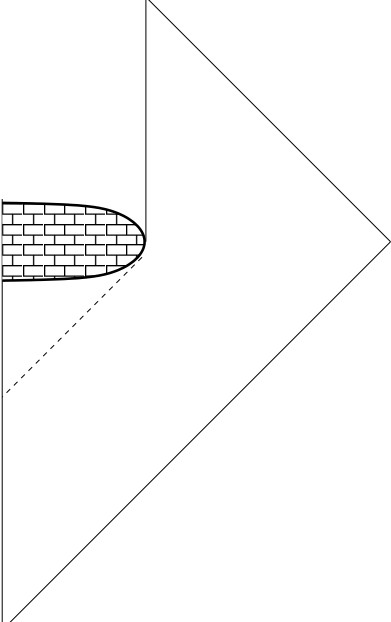

**Figure 6.** The Euclidean region as a final boundary on which future data are posed and not evolved further.

### 3.6. Unexpected Relationships with Other Approaches

The final-state scenario already shows that models that take signature change seriously lead to unexpected similarities between what has been suggested in loop quantum gravity and other approaches. The absencce of bounces in cosmology or in black holes is also surprisingly consistent with the no-transition principle extracted in [97] from a gauge-gravity correspondence, which does not permit exchanges between independent quantum field theories on a boundary. Although signature change is independent of boundary components, it prevents the same kind of transitions as discussed in [97]. In [8], by contrast, a contradiction between bouncing black holes and the no-transition principle has been observed, claiming that this is possible because in loop models "one works directly in the bulk" (emphasis in [8]). However, a transition is possible in the model of [8] only because this paper failed to capture the correct bulk geometry, or rather to implement any consistent bulk geometry as per the covariance conditions given in Section 2.1.

The possibility of signature change (or mixed-type problems) in models of loop quantum gravity is similar to certain higher-curvature theories such as Einstein-dilaton Gauss–Bonnet gravity [98,99]; see also [100–103] for a general analysis in Horndeski theories [67]. In models of loop quantum gravity, signature change does not require higher-curvature terms and is a result of modified space-time structures, not just of modified dynamics. Nevertheless, there are interesting relationships between implications for black holes if signature change can always be confined to regions hidden behind horizons.

In the context of quantum gravity, different versions of signature change have been found in a variety of approaches, including minisuperspace models [104], string theory [105], matrix models [106–109], and causal dynamical triangulations [110].

### 3.7. Avoiding Signature Change in Models of Loop Quantum Gravity

It is sometimes possible to construct holonomy modifications of certain models without implying signature change. For instance, the Hamiltonian constraint expressed using complex connections instead of the momenta in (15) has a different appearance of spatial derivatives of metric components. It can be consistently modified in ways different from (25), in particular such that even the modified constraints obey (27) with $\beta = 1$ [111–113]. However, these modifications are not generic because a complete effective description to the same order of derivatives would include spatial derivative

terms as in (15), and signature-changing brackets would result [78,114]. Working with generic modifications, the required relation (31) implies signature change for any function $f_1$ that has a local maximum. Therefore, signature change cannot be removed by familiar regularization options in loop quantum gravity, in particular by choosing an $SU(2)$-representation to represent holonomies. Any such representation leads to a bounded and periodic function $f_1$ which therefore has a local maximum. Representation choices in the context of deformed hypersurface-deformation brackets have been considered in [115,116].

Another possibility to avoid signature change can be found if one allows for non-bouncing solutions in loop quantum cosmology [41], at least in the context of cosmological perturbations. The bracket (27) with $\beta$ evaluated on a non-bouncing background solution is then still modified, but $\beta > 0$ throughout the high-density phase. A single Riemannian geometry described by an effective line element (39) then exists for the entire transition. Since these solutions are non-bouncing, they reach zero volume and may trigger a curvature singularity. However, the dynamics is modified compared with classical cosmological models and could remain non-singular, but this possibility has not been explored yet for the solutions described in [41]. As supporting evidence for non-singular behavior, it is known that the equations of loop quantum cosmology can remain non-singular even in a transition through zero volume [117]. What is at present unclear is whether such a transition can be described in an effective model suitable for the geometry of a black hole.

Signature change—or, more generally, a deformation of the hypersurface-deformation brackets—is derived in models of loop quantum gravity from one specific quantum effect, given by holonomy modifications in the Hamiltonian constraint. In addition, in a perturbative treatment of any interacting theory one would expect a large number of quantum corrections from loop diagrams which, in the case of gravity, usually come in the form of higher-curvature corrections [118,119]. In a canonical treatment, the corresponding higher-derivative terms can be seen to be implied by quantum back-reaction of fluctuations and higher moments on the expectation values of basic variables [28,29,120]. Based on general algebraic properties of moments, derived from the quantum commutator, it can be shown in general terms [121] that quantum back-reaction on its own cannot produce modified structure functions as in (27), and therefore does not interfere with any such modification implied by other effects, such as holonomies. This result, which is consistent with the fact that no higher-curvature effective action of general relativity modifies the hypersurface-deformation brackets [79], shows that signature change is robust under the inclusion of quantum back-reaction or corrections from loop diagrams. It is at present unknown how non-perturbative effects might affect signature change or, more basically, the various effective formulations currently used in all models of loop quantum gravity. The issue of signature change (and how it could be avoided) provides strong motivation for loop quantum gravity to focus on studying the challenging question of how to understand non-perturbative, background-independent quantum effects in a space-time picture.

Even if signature change could be avoided, its possibility is of conceptual importance because it shows that proposed models that do not address (or even explicitly violate) the covariance conditions spelled out in Section 2.1 cannot be considered "first approximations" to some complicated full theory of quantum gravity, as sometimes suggested. The possibility of signature change shows that a failure to address the covariance problem can have drastic consequences because it leads one to misinterpret the causal structure consistent with one's proposed equations. By considering the covariance problem, one includes strong consistency conditions that can rule out certain proposals, as shown here.

## 4. Conclusions

A majority of previously proposed scenarios for evaporating black holes in models of loop quantum gravity suffer from several severe problems. In particular, they use equations that can be shown explicitly to violate general covariance, and they are based on several crucial assumptions which have never been demonstrated in this setting. An example of the latter is the postulate that the interior of a black hole may bounce and open up again such that it is causally connected to the former

exterior. While bounces can be motivated by more tractable cosmological models, using the well-known homogeneous slicing of the Schwarzschild interior, showing that the interior reconnects to an exterior space-time requires inhomogeneous models. In inhomogeneous models, however, the equations of loop quantum gravity are much more complicated and cannot be solved yet in any controlled way. If models are used that insert potential loop effects into classical equations, there is a large number of ambiguities as well as problems with covariance.

An interesting suggestion made in [8] initially indicated that tractable homogeneous models could be applied even in the inhomogeneous exterior, provided they are based on timelike slicings applied to static solutions. However, loop modifications in the timelike slicing cannot be compatible with general covariance [2,4]. Instead of producing a viable model of quantum black holes, the proposal of [8] provided a crucial step in a demonstration that models of loop quantum gravity violate general covariance.

Nevertheless, models of loop quantum gravity may be consistent provided they incorporate a generalized version of covariance in which classical slicing independence is replaced by a new quantum symmetry. In certain cases, field redefinitions can be used to map the variables of such a model to an effective metric which can be represented in the standard form of Riemannian geometry and is consistent with slicing independence [81]. However, it remains unknown whether such transformations are always possible. The precise geometrical nature of generalized covariance is therefore unclear, complicating detailed analyses of black-hole models in loop quantum gravity.

One common implication of generalized covariance in models of loop quantum gravity is dynamical signature change, which usually takes place at large curvature. Even while a precise geometrical description remains unknown, corresponding consequences can be analyzed in general terms because signature change implies a characteristic form of well-posed initial-boundary value problems. When Euclidean regions are located to the future of a Lorentzian low-curvature region, the boundary data they require for well-posed problems amount to future data from the perspective of the Lorentzian region. They imply the danger of violating deterministic behavior even for low-curvature observers who never directly experience strong quantum-gravity effects. This implication is sufficient to rule out the idea of Planck stars in models of loop quantum gravity, given by bouncing black-hole interiors that reconnect with the former exterior and may become visible. Two consistent options for deterministic scenarios are given by an orphan universe in which a baby universe splits up in a causally disconnected way, and a final-state scenario in which the interior is not extended but the classical singularity is replaced by a final condition.

These scenarios of evaporating black holes have been described in Section 3. A consistent scenario for a vacuum black hole, extending the classical Kruskal space-time, is shown in Figure 7, based on the following two statements: (1) Interiors are non-singular, but they do not bounce because evolution is blocked by Euclidean regions. (2) Future interiors cannot reconnect with past exteriors because the boundary data required in Euclidean regions would then lead to violations of deterministic behavior at low curvature. The scenario shown in Figure 7 is parsimonious: it is consistent with low-curvature determinism, and it does not require a complicated dynamical analysis to see whether solutions of loop quantum gravity make it possible for a bouncing interior to reconnect with a former exterior. The space(-time) required for such a scenario can instead be patched together from non-singular interiors and almost-classical low-curvature exteriors, with a resulting global structure that follows from general consistency requirements. Even though global properties are derived, they are implied by a careful local analysis of covariance in field equations.

In this scenario, future interiors are unobservable in past exteriors. The model is therefore not directly testable by observations, but it is preferred by a combination of internal consistency relations, including covariance and low-curvature determinism. Based on these conditions, the scenario shown in Figure 7 presents the limit of what can at present be claimed in models of loop quantum gravity. Future studies will have to show whether its limitations to evolution implied by signature change

are a consequence of currently available effective descriptions, required for any detailed analysis, or whether they hold even in a full theory of loop quantum gravity.

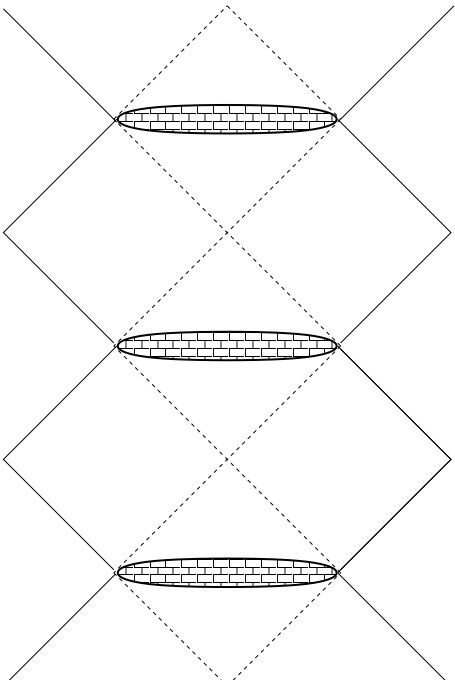

**Figure 7.** Quantum extension of a Kruskal black hole. While curvature singularities may be removed in models of loop quantum gravity, they are replaced by regions of Euclidean space through which evolution is impossible. The extended space(-time) has unique solutions provided well-posed initial-boundary value problems are used for mixed-type equations. Lorentzian regions have finite boundaries at their transition hypersurfaces to Euclidean regions, which are non-singular but require boundary values that are interpreted as future data from the Lorentzian perspective. The extension shown here is derived from covariance together with consistency with deterministic behavior at low curvature.

**Funding:** This research was funded by NSF grant number PHY-1912168.

**Conflicts of Interest:** The author declares no conflict of interest.

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
