# Peer review of "Black-Hole Models in Loop Quantum Gravity"

_universe, doi:10.3390/universe6080125_

Round 1

Reviewer 1 Report

The quality of this work does not meet the standards of Classical and Quantum Gravity.  The introduction and the motivation shows confusion about the general form of the classical limit of a quantum theory.  The argument that there cannot be an effective classical-like metric describing a quantum transition does not hold. The author appear to be focused on a questionable result of his in the canonical formalism and fails to see that his way of searching for a theory is not the only possible one.  

Author Response

I appreciate the effort you put into this review. Unfortunately, however, your
comments are imprecise to a degree that makes them useless. The questions
below will help you to clarify your viewpoint. In the future, please remember
to support all your claims by justifications, in particular when you are
accusing a fellow scientist of basic misunderstandings. Note also that you
misidentified the journal in which my submission is under review. It is not
Classical and Quantum Gravity but Universe.

1. Where exactly do you see "confusion about the general form of the classical
limit of a quantum theory"? Notice that there is a difference between a
classical limit and a semiclassical description. My review is discussing the
latter, while your comment is about the former. In case you are referring to
the classical limit I mention in condition (ii.b'), page 4, notice that this
condition is to be imposed not in isolation but in conjunction with condition
(ii.a). While condition (ii.b') is for the classical limit, condition (ii.a)
must hold for a semiclassical description where hbar is not zero. Therefore,
just having the correct classical limit does not imply covariance. I have
added such a comment at the end of the paragraph following condition (ii.b').

2. Why do you think that "The argument that there cannot be an effective
classical-like metric describing a quantum transition does not hold." Notice
that I am making this claim only in specific situations and not with such a
generality as implied by your comment. In my review I have produced not just
an argument but explicit examples of models in which a metric is not
available.

3. Which result specifically do you have in mind when you say that "The author
appear to be focused on a questionable result of his in the canonical
formalism", and why do you think that it is questionable? My review focuses on
the broad question of covariance in different formulations, mostly canonical
ones, rather than on any single result.

4. Why do you think that I fail "to see that [my] way of searching for a
theory is not the only possible one"? Did you notice that I am addressing four
different approaches in this review, including Rovelli et al.'s Planck stars,
the recent Ashtekar-Olmedo-Singh construction, and Gambini-Pullin models in
addition to some of my own results?

Reviewer 2 Report

This is a very interesting, informative and well-written paper. I recommend publication.

I just have one question, which I leave up to the author whether he wants to take it into account. In the first paragraph of 2.3.2, it is stated that gauge fixing before quantization leads to questionable physics. I didn't know that. It is routinely done e.g. in electrodynamics. What exactly goes wrong with in the case of the black hole models? Fixing the gauge before quantization is also akin to reduced phase space quantization and that is often promoted as the preferred way to quantize.

A typo: line 421: "indpendence"

Author Response

Good point. I have appended "unless it can be shown that a covariant
quantization of this kind exists. Without an explicit demonstration of gauge
invariance none of the conditions..." to the statement in question. In quantum
electrodynamics one knows that gauge-invariant quantizations exist, and can
therefore derive their properties more easily by working with a gauge
fixing. However, constructing a new quantum model based only on a gauge-fixed
description does not guarantee consistency.

Thank you also for pointing out the typo.

Reviewer 3 Report

In this review, the author summarizes and deals with serious problems presented in loop quantum gravity-inspired modifications (governed by a parameter) of the Poisson brackets of general relativity that aim to describe a “quantum black hole”. Such deformations are primarily used in loop quantum cosmology and are usually motivated by the application of holonomy operators in the theory, which justifies the deformation to be of periodic nature.

The major issues are related to a lack of covariance of the black hole solutions (or at least the lack of examination of the covariance), i.e., the existence of coordinate-transformed metric components that are solutions of transformed field equations. And also the preservation of the tensorial law that transforms these metric components. Which seem to be reasonable conditions.

The author demonstrates that the cases under study do not satisfy all of those requirements, however at least one of them [R. Gambini and J. Pullin, PRL 110, 211301 (2013)] satisfies the majority of the covariant requirements and a weaker version of the tensorial rule.

Despite being able the satisfy a new set of covariant postulates, that justify describing this approach as a “deformed general relativity”, some other issues are raised.

First, the author claims that since one cannot construct a two-indexes object from the solutions of the field equations (which is supposed to be the space-time metric for which one constructs a line element) that transforms as a tensor (only when the deformation parameter is negligible or in special cases detailed in the paper) implies that one is not dealing with Riemannian geometry at this level.

“As a matter of fact, it seems that such approach not only cannot be described by the language of Riemannian geometry, but also of non-Riemannian geometries, like those with non-metricity and torsion fields, since the intrinsic/tensorial language cannot be applied (at least in the general case).

So, this is something that needs to be pointed out.”

This is a very interesting issue, that represents a real challenge for the researchers of this area.

In second place, there emerges a signature oscillation from Lorentzian to Euclidean that is also curious, since it affects the kind of initial value formulation of the theory and seems to break its deterministic nature. Also, the singularity theorems can no longer be used in this situation because there is no causal structure in the Euclidean regime and the tools and results from differential topology used in theorems no longer hold.

This property is related to the choice of deforming function f_1 and consequently the function beta that modifies the brackets of the hamiltonian constraint (that fact that it is periodic). However, as mentioned in section 3.7, it is possible to avoid a signature change, which currently is a research topic.

“I missed specifically the effect that other choices of regularization schemes for different spin representations may have on this approach (see for instance [S. Brahma and M. Ronco, PLB 778, 184 (2018), arXiv:[1801.09417]] and references therein, where different kinds of deformation functions are analysed). I.e., if these other possibilites are already examined in this context and/or a discussion if this represents an interesting alternative.”

Finally, I just would like to point out the unusual analogy presented in line 456, where criticisms to signature change are compared to anti-vaxxer arguments. I confess that I feel a bit uncomfortable reading such harsh (maybe aggressive) observation. But I think that requiring specific modifications on the way the author choses to present the state of the art of this area goes beyond my role as referee. In any case, I just would like to express my surprise with this choice of words.

In summary, the review paper discusses updated topics related to quantum black holes from que loop quantum gravity perspective showing their weak and strong points, and helps to pave the way to future developments. Therefore I recommend the paper for publication after the two minor issues quoted above (italic) are addressed.

Author Response

Thank you for your detailed evaluation. I have inserted the following passage
in the introduction in order to address your first suggestion: "The resulting
non-Riemannian structure would have to deviate from standard constructions in
very basic ways because it is modified even in properties of generic tensorial
objects encoded by the tensor-transformation law (which is replaced by gauge
transformations in a canonical formulation). Therefore, deformed geometrical
properties cannot be captured completely by well-known classical ingredients
such as torsion or non-metricity." (lines 68-73). The second half of the first
paragraph in Section 3.7 (page 21) now contains a few statements about higher
spin representations. Briefly, as long as the modification functions remain
periodic and bounded, they have local maxima at curvature values where
signature change takes place. Thank you for commenting on my statement about
anti-vaxxers, which was meant to be humorous. (Seeing this objection again and
again, long after it has been discredited, does feel comical.) But I see how
it can be misunderstood, and therefore removed it.

Reviewer 4 Report

The author reviews in great detail the state-of-the-art of black-holes models arising from or motivated by loop quantum gravity. The review is very detailed, precise, and objective.

I only have a few (very minor) comments and some questions:

1) In the introduction, the sentence "Recently, it has been shown by direct calculations that all current loop-inspired black-hole models of bounce form violate general covariance and therefore fail to describe space-time effects in any meaningful way." should be followed by one or more references to the article(s) showing this fact.

2) The author state that "If the Hamiltonian constraint is modified, gauge transformations may be non-classical". After this sentence (and perhaps also in the conclusions) it would be nice to have a better discussion on the possible fundamental quantum symmetries of the theory. In particular, discussing any possible new (reduced) symmetry, the possible violation of Lorentz invariance, and its consequences.

3) In relation to the previous point, if the classical symmetries of the theory are broken, how can loop quantum gravity be background independent (in the classical sense)? The author should comment on this point either in the introduction or in the conclusions.

4) In Section 2.1 the author discusses an LQG-inspired model based on the Hayward metric, and its compatibility with general covariance. However, this discussion is misleading since the Hayward metric can actually be obtained as a solution to modified Einstein equations (which are covariant). The author should better clarify this point.

5) In the conclusions, it would be nice to have a discussion about how stable is the prediction of LQG that at high energies there should be a dynamical signature change that makes the spacetime Euclidean. Is this a very general finding of LQG? Or, this could change in case some assumptions are modified? Which assumptions would need to be modified to avoid the signature change? Also, a particularly pressing issue is that the resummation of all quantum effects and renormalization effects could generate, at the level of the action, additional higher-derivative terms. Can the author comment about this point and also discuss its possible implications for the predictions of LQG and for the possibility of signature-changing spacetimes?

Author Response

Thank your for your evaluation and helpful suggestions.

1. I have inserted four references at this place, and also took the
opportunity to expand this discussion by adding "It is important to note that
these results were obtained directly within the proposed models. They are
therefore independent of any difference between approaches that have been put
forward in order to formulate inhomogeneous models of loop quantum gravity,
such as hybrid models [], the dressed-metric approach [], partial
Abelianization in spherically symmetric models [], or using timelike
homogeneous slices in static spherically symmetric space-times [], just to
name those to which the recent no-go results about covariance directly apply."
where "[]" indicate citations to papers that have been found to be in conflict
with covariance.

2. and 3. I prefer to address these questions by an extended discussion in the
introduction, where I now have the additional comments "Deformation of the
algebra, as opposed to violation of some gauge transformations as found in
most of the approaches mentioned in the preceding paragraph, makes sure that
the theory remains background independent in the sense that transformations
remove the same number of gauge degrees of freedom as in the classical
theory. However, as a consequence, space-time is rendered non-Riemannian ---
unless field redefinitions are applied in certain cases --- and at present no
universal non-Riemannian space-time geometry has been identified that could
describe all the deformation effects.", "Such a deformed theory is still
predictive in principle because it has a consistent canonical formulation that
defines unambiguous observables. But the relationship between gauge invariance
and slicing independence (according to a suitable space-time structure) is
less obvious than in the classical limit, complicating interpretations of
space-time effects such as black holes." and "Possible space-time structures
consistent with deformed symmetries are still being explored. For instance, it
is known that deformations of the form implied by effects from loop quantum
gravity are different from modified coordinate transformations found in
non-commutative [] or multifractional geometry []. There are some
relationships with deformed Poincare symmetries in suitable limits in which
a Minkowski background is obtained [], which confirms that Lorentz
transformations are not necessarily violated, but the Minkowski reduction of
general space-time transformations is rather strong and cannot be sufficient
for a complete understanding of deformed space-time structures. As we will
show here, what is known about the resulting consistent space-time structures
suggests a markedly different scenario of non-singular black holes, compared
with bounce-based black holes." between lines 58 and 82.

4. I agree that my previous reference to Hayward's metric may have been
misleading. I have extended this reference to a longer discussion, "The
specific function (2) was first proposed in [], where it was also shown that
it may be obtained as a solution of a covariant theory, such as general
relativity with a suitable stress-energy tensor that falls off quickly with
increasing r. The question we pose here is more fundamental and works in the
opposite direction: Starting with a specific modified theory or at least a set
of model equations that are not gauge-fixed so as to keep space-time
properties accessible, is it possible to express its solutions in the form of
Riemannian line elements consistent with gauge transformations."

5. Also here, I prefer a slightly different place for such a discussion, not
in the Conclusions but close to them in Section 3.7. The next-to-last
paragraph of this section is new and addresses the question: "Signature change
--- or, more generally, a deformation of the hypersurface-deformation brackets
--- is derived in models of loop quantum gravity from one specific quantum
effect, given by holonomy modifications in the Hamiltonian constraint. In
addition, in a perturbative treatment of any interacting theory one would
expect a large number of quantum corrections from loop diagrams which, in the
case of gravity, usually come in the form of higher-curvature corrections
[]. In a canonical treatment, the corresponding higher-derivative terms can be
seen to be implied by quantum back-reaction of fluctuations and higher moments
on the expectation values of basic variables []. Based on general algebraic
properties of moments, derived from the quantum commutator, it can be shown in
general terms [] that quantum back-reaction on its own cannot produce modified
structure functions as in (27), and therefore does not interfere with any such
modification implied by other effects, such as holonomies. This result, which
is consistent with the fact that no higher-curvature effective action of
general relativity modifies the hypersurface-deformation brackets [], shows
that signature change is robust under the inclusion of quantum back-reaction
or corrections from loop diagrams. It is at present unknown how
non-perturbative effects might affect signature change or, more basically, the
various effective formulations currently used in all models of loop quantum
gravity. The issue of signature change (and how it could be avoided) provides
strong motivation for loop quantum gravity to focus on studying the
challenging question of how to understand non-perturbative,
background-independent quantum effects in a space-time picture."

Round 2

Reviewer 1 Report

This article is wrong and misleading.   I was amused to see the report of the second referee to state "I just have one question, which I leave up to the author whether he wants to take it into account. In the first paragraph of 2.3.2, it is stated that gauge fixing before quantization leads to questionable physics. I didn't know that. It is routinely done e.g. in electrodynamics."  This summarises well the entire issue.   The referee "did not know that" because that is not true.  The author has his own (questionable) approach to quantisation and judges everything else wrong because it does not conform to his own idiosyncratic quantisation procedures.